# The Measurement of Hazardous Biogenic Amines in Non-Alcoholic Beers: Efficient and Applicable Miniaturized Electro-Membrane Extraction Joined to Gas Chromatography-Mass Spectrometry

**DOI:** 10.3390/foods12061141

**Published:** 2023-03-08

**Authors:** Marzieh Kamankesh, Fatemeh Barzegar, Nabi Shariatifar, Abdorreza Mohammadi

**Affiliations:** 1Food Safety Research Center (Salt), Semnan University of Medical Sciences, Semnan 35147-99442, Iran; 2School of Pharmacy, Semnan University of Medical Sciences, Semnan 35147-99442, Iran; 3Department of Food Science and Technology, Faculty of Nutrition Science, Food Science and Technology/National Nutrition and Food Technology Research Institute, Shahid Beheshti University of Medical Sciences, Tehran 198396-3113, Iran; 4Department of Environmental Health, Food Safety Division, School of Public Health, Tehran University of Medical Sciences, Tehran 14176-13151, Iran; 5Food Safety Research Center, Shahid Beheshti University of Medical Sciences, Tehran 198396-3113, Iran

**Keywords:** beer, biogenic amines, electro-membrane extraction, gas chromatography-mass spectrometry

## Abstract

The determination of biogenic amines (BAs) as serious food contaminants and chemical indicators of unwanted microbial contamination or deficient processing conditions in non-alcoholic beers is of great interest for the beverage industries. In the present investigation, the combination of hollow fiber-electro-membrane extraction (HF-EME) and dispersive liquid–liquid microextraction (DLLME) followed by gas chromatography-mass spectrometry (GC/MS) was applied for the analysis of histamine, putrescine, tyramine, cadaverine in non-alcoholic beers. EME is fundamentally based on the electrostatic attraction, diffusion and solvability of analytes in a selected acceptor phase. This membrane-based extraction technique promoted selectivity and the enrichment factor. The DLLME process reduced the volumes of organic solvents and make the coupling of HF-EME to the CG/MS conceivable. The leading variables, which have a great effect on extraction recovery, were optimized. The relative standard deviation was achieved between 4.9 and 7.0%. The recoveries were between 94% and 98%. The limit of detection and limit of quantification were found to be 0.92–0.98 ng mL^−1^ and 3.03–3.23 ng mL^−1^, respectively. The enrichment factor was calculated in the range 36–41. The achievements revealed that putrescine and tyramine, with concentrations of 3.87 and 2.33 µg g^−1^, were at the highest concentration in non-alcoholic beers. This offered method with great benefits could help beverage industries to monitor the concentration of BAs in beers and control them.

## 1. Introduction

Biogenic amines (BAs) are identified as low molecular nitrogenous compounds that are produced by enzymatic reactions, such as the decarboxylation of free amino acids and corresponding precursors or by the amination or transamination of ketones and aldehydes [1]. These organic compounds divide into aliphatic (cadaverine, putrescine, spermine, spermidine, agmatine), aromatic (tyramine, 2-phenylethylamine) and heterocyclic (histamine, tryptamine) groups [2]. BAs naturally exist in human body as signaling molecules and play physiological roles. Scientific-based evidence has proved that the excess consumption of BAs triggers allergic reactions, hypotension, gastrointestinal disorders, headaches, kidney damage, anaphylactic shock and even death [1,3]. Generally, 100 mg/kg or 100 mg/L of BAs is regarded as a safe dose for most consumers [4]. BAs are also considered as valuable indicators for the evaluation of food quality [5,6]. The exogenous sources of BAs are fermented food, meat products, cheese, fishery products, fruits, vegetables and beverages [2,7]. For instance, beer is one of the popular beverages containing threatening BA compounds.

Beer is deemed as the oldest favorable beverage in human life. The unique aroma and taste of beer have been the leading factors of its popularity in different eras [8,9,10]. Raising awareness about the health benefits of non-alcoholic and low-alcohol beers rapidly develops the market of these beverages around the world. Non-alcoholic beer is labeled with various names such as “small beer”, “alcohol-free beer”, “near beer” and “de-alcoholized beer” and contains 0.00–0.50% alcohol [11]. As a matter of fact, the de-alcoholization procedure makes the consumption of beer with health-promoting components possible, including phenolic compounds and specific vitamins and minerals, without extra alcohol intake. While the presence of BAs in this kind of beer is a threatening factor, the amount BAs in beer is not related to its alcohol content. It was pointed out that non-alcoholic beers do not have a remarkably lower concentration of BAs than alcoholic beers [10]. BAs are originated from the fermentation process and can also formed by microorganisms existing in malt, hops, barley and wort. The level of BAs in beer is approximately variable and relies on different factors such as the brewing method, raw materials quality, storage condition and hygiene. It was reported that histamine, putrescine, agmatine, cadaverine, spermine, spermidine, 2-phenylethylamine and tyramine are the most common BAs detected in beer [4]. It is required to consistently monitor BAs in non-alcoholic beer due to quality and safety assessments.

Several attempts have been made to analyze BAs in various food products by using chromatographic instruments, including gas chromatography-mass spectrometry (GC-MS) [12,13] and high performance liquid chromatography [14,15]. Other analytical techniques, including capillary electrophoresis [16], liquid chromatography tandem-mass spectrometry (LC-MS/MS) [17], and ultra-high performance liquid chromatography [18] are also applied for the measurement of BAs.

It has long been established that the straightforward analysis of samples without the sample preparation process is not possible. Therefore, a sample pretreatment and extraction method is required to reduce matrix interferences, increase sensitivity and facilitate analyte release from intricate food tissue [19]. The application of classical extraction methods was dramatically reduced, owing to the requirement of high amounts of toxic solvents, long extraction time, high costs and the high risk of analyte losses [20]. Therefore, these traditional methods gave way to a new microextraction procedure. Liquid phase microextraction (LPME) is the new generation of traditional liquid–liquid extraction (LLE), which is based totally on the equal dispensation of the specified analytes between the sample solution and extraction phase. The volume of the extraction solvent is miniaturized and the recovery factor is potentiated [21,22]. Dispersive liquid–liquid extraction (DLLME) is one of the subsets of LPME that can efficiently extract various analytes from food. In this method, the low volume of extraction and disperser solvent are swiftly injected into an aqueous sample solution. The dispersive solvent acts as an intermediary solvent and distributes the small particles of the extraction solvent throughout the aqueous phase and increases the interaction between the extraction phase and aqueous sample solution [23]. Recently, potent electro-membrane extraction has attracted researchers’ attention for the efficient separation of ionizable analytes and sample solutions. This emerging technique was introduced in 2006 and is based on the electromigration of charged compounds by an electric current. The ionized species are transferred from the donor phase (sample solution) across a supported liquid membrane (SLM) into an acceptor phase (extraction solvent) by an applied electrical force during a specific time. This method is particularly useful due to the high enrichment factor and selectivity [24,25]. Hollow fiber electro-membrane extraction (HF-EME) is one of the EME setups that dexterously was designed to improve recovery and enrichment [24]. The porous polypropylene hollow fiber acts as a holder for the acceptor phase. A thin layer of SLM is also formed on the inner wall of the HF, which is considered as an SLM supporter. The level and the time of exerted voltage, the SLM type, the pH of the acceptor and the donor phase are the crucial variables that substantially influence the electro-separation of target analytes [26]. A recent investigation in 2023 used conductive cloth-assisted electromediated extraction for the determination of putrescine, cadaverine, phenylethylamine, tyramine and tryptamine in orange and apple juices [27]. The combination of both DLLME and HF-EME contributes to the potentiation of the extraction process and improves the isolation of target compounds. The value of this new design of extraction procedure has been recognized by the considerable reduction of toxic organic solvents applied in DLLME. This modification also enhances the safety aspect of the clean-up process and promotes sensitivity and the enrichment factor [21].

In the present experiment, the most-common and problematic Bas, such as histamine, putrescine, cadaverine and tyramine, were analyzed in non-alcoholic beer using the powerful EME-μ DLLME technique joined with GC-MS. The effects of the main factors on the extraction mechanism have been opted and optimized by applying one variable at a time (OVAT). The validity of the offered method was confirmed in the following, and the obtained consequences corroborated the accuracy and applicability of this emerging technique for the measurement BAs in non-alcoholic beers.

## 2. Materials and Methods

### 2.1. Chemicals and Reagents

Tyramine (97%), histamine (98%), putrescine (99%), and cadaverine (99%) as main standards were individually ordered from Fluka (Buchs, Switzerland). Hydrochloric acid (37%), 1-octanol, acetonitrile, sodium chloride, acetone, potassium hydroxide (KOH), ethanol, methanol, 2-nitrophenyl octyle ether (NPOE), dipotassium hydrogen phosphate (K_2_HPO_4_), potassium hexaferrocyanide and zinc acetate were purchased from Merck (Darmstadt, Germany). Isobutyl chloroformate was obtained from Sigma-Aldrich (St. Louis, MO, USA). Ten non-alcoholic beer samples were randomly purchased from different supermarkets (Tehran, Iran).

### 2.2. Stock Standards Preparation

The mixed stock standard solutions of the abovementioned biogenic amines at concentrations of 2000 mg L^−1^ were created in HCl 0.1 mol L^−1^. HCl (0.1 mol L^−1^) was applied for the dilution of the stock standard solution to produce the working standard solution in various required concentrations. In order to produce phosphate buffer solution (0.5 mol L^−1^, pH = 12), 43.5 gram of K_2_HPO_4_ and KOH at a concentration of 2.0 mol L^−1^ were completely mixed. Carrez (I) was also obtained by dissolving 10.6 gram of potassium hexaferrocyanide in 100 mL of distilled water. For the preparation of Carrez (II), 21.9 gram of zinc acetate was added to 3 mL acetic acid, and we regulated the final volume to 100 mL using distilled water. All of the obtained solutions were stored at 4 °C.

### 2.3. Equipment and Chromatography Analysis

The electro-membrane system was comprised of two simple platinum electrodes with diameters of 0.5 mm (anode) and 0.2 mm (cathode), which were obtained from Pars Pelatine (Tehran, Iran). The porous PPQ3/2 polypropylene HF with a thickness of 200 μm, pore size of 0.2 μm, and internal diameter of 0.6 mm, was applied as an acceptor phase holder and SLM supporter (Membrana, Wuppertal, Germany). The constant voltage in the limit of 0–600 V was provided by power supply model 8760T3 (Paya Pajoohesh Pars, Tehran, Iran). The mixed phases were isolated by centrifuge (Hettich, Tuttlingen, German). A magnetic stirrer (model 301) was also employed for stirring during extraction (Heidolph, Kelheim, Germany).

After the extraction procedure, the chromatographic analysis was conducted using GC-MS, model 7890A (Agilent Company, Palo Alto, CA, USA). This analytical system is equipped with a triple-axis detector, fitted with a split/split-less injector, and joined with a 5975C inert MSD network mass-selective detector. The extracted biogenic amines were separated by an HP-5 MS capillary column containing 5% phenyl siloxane/95% methylpolyorganosiloxan (length: 30 m, inner diameter: 0.25 mm, film thickness: 0.25 μm). The oven temperature was adjusted as follows: 190 °C for 5 min; raised to 250 °C at 7 °C/min and kept for 1 min; and increased to 300 °C at 20 °C/min and held for 10 min. The carrier gas was helium, with a flow rate of 1 mL/min. The injector temperature was 280 °C and a split ratio of 1:50 was applied. The selected ion-monitoring acquisition mode was utilized for the analysis of the BAs. The mass to charge (m/z) for histamine, putrescine, tyramine and cadaverine was 194, 170, 107 and 130, respectively.

### 2.4. Initial Sample Preparation

Five milliliters of non-alcoholic beer sample was poured in the test tubes and spiked with the mixed standard solution of BAs at a concentration of 1000 ng/g. Then, 12 mL of HCl 0.060 mol/L was added and completely mixed for 1 min. The usage purpose of HCl solution is pH regulation, for the extraction of BAs from the sample matrices. Then, 0.5 mL of Carrez I and 0.5 mL of Carrez II were poured and mixed for 1 min. The Carrez solutions were applied for the removal of interferences and protein precipitation. Afterward, the obtained solution was centrifuged at 9000 rpm for 5 min, and the upper solution phase was separated and used for the next step.

### 2.5. Electro-Membrane and Microextraction Process

The final solution obtained from the previous step was poured into a glass vessel with a height of 6 cm and an inner diameter of 2.5 mm. A piece of fiber was cut to 3.5 cm and soaked in the NPOE, as a suitable SLM. Afterward, the extra NPOE was slowly wiped away with a micro-syringe, and 10 μL of HCl solution (0.1 mol L^−1^) was injected into the fiber, which acts as an acceptor phase. The bottom of the HF was completely sealed with a thick aluminum end-cap to prevent leakage. The cathode was gently entered into the HF and the anode was placed into the aqueous sample solution. The prepared EME setup was placed on the magnetic stirrer and the speed of rotation was adjusted to 650 rpm. The power supply was connected and exerted 55 V as the electrical potential for 20 min. After the extraction process, the HCl solution saturated with target BAs was removed and mixed with 1 mL of phosphate buffer (pH = 12) in a micro-tube. The provided solution was fully shaken and 2.5 μL of isobutyl chloroformate (as a proper derivatization reagent) was added. The next step was continued by DLLME procedure. Then, 0.02 gram of NaCl was added to the achieved solution and properly mixed. Subsequently, 20 μL of 1-octanol as an extraction solvent and 100 μL of acetonitrile as a dispersive solvent were instantly injected into the micro-tube and properly shaken to achieve the cloudy phase. After centrifugation, the upper phase was injected into the GC-MS for further analysis.

### 2.6. Statistical Analysis

The critical variables substantially affected both extraction methods selected and were optimized by one variable at a time to attain the highest response. The level of applied potential (20–100 v), the time of applied potential (10–50 min), the speed of rotation (300–700) and the ion balance (0–1.2) were chosen as the most effective EME variables. The volume of dispersive (300–800 µL) and extracting solvent (40–100 µL), salt percent (0–20%) and pH (2–11) level were also selected as major factors that influenced the DLLME mechanism. The number of replicates for each experiment was three, and the average response was considered as the final response. *p* ≤ 0.05 was chosen as statistically significant for the test results. In the optimization stage, one of the variables was changed and the other variables remained constant. The NPOE was also selected as the SLM. The average peak area of BAs indicated the optimization response, to assess the extraction efficiency. After the optimization of effective factors, the merit figures were calculated and reported, and the standard addition (SA) test was applied for the validation of the results.

## 3. Results and Discussion

The selected factors impressed on HF-EME method were initially optimized. Thereafter, the main variables, which have far-reaching effect on the DLLME mechanism, were also optimized. There is an unequivocal relationship between SLM type and HF-EME performance. The accurate selection of the SLM could guarantee the stability and the efficiency of the EME system. Investigations has consistently revealed that NPOE is the best organic solvent for playing the role of the SLM. This non-volatile solvent is extremely stable at a high voltage level and is also immiscible in water, so could not leak into the aqueous sample solution. NPOE has a high permeability for the mass transfer of ionic species and is compatible with HF [26]. Accordingly, this organic solvent was chosen as a proper SLM.

### 3.1. Optimization of HF-EME Technique

#### 3.1.1. Speed of Rotation

The stirring rate could be a contributing factor in BA extraction. HF-EME is carried out under a stirring arrangement. The induced convection to a sample solution simplifies the ionized species mass transfer toward the SLM. Moreover, agitation decreases the thickness of the boundary layer at the sample/SLM junction for improving mass transfer [26]. In the current survey, three levels of rotation speed (300, 500, 700 rpm) were tested, and 700 rpm was achieved as the optimal value (Figure 1). It was highlighted that a higher rotation speed brings about bubble formation and cause to stop the extraction process.

#### 3.1.2. Donor Phase and Acceptor Phase Nature (Ion Balance)

The nature of the donor and acceptor phases was explained in light of the ion balance factor. Ion balance is known as the ratio of the H^+^ level in the donor phase to the H^+^ level in the acceptor phase [28]. The BA extraction yield would be upgraded with an increase in H ^+^ amount in the acceptor solution or an ion balance reduction due to the basic characteristics of BAs. Six values of ion balance factor (0, 0.2, 0.4, 0.6, 0.8, 1) were studied to evaluate the extraction efficiency. The considerable concentration of H^+^ in the donor phase stimulated the competition between BAs^+^ and hydrogen ions for migration to the SLM. This means that the electrical layer thickness was enhanced, and consequently the analyte transfer to the acceptor solution ceased. The line graph in Figure 1 exhibits that 0 was the optimum value for the ion balance factor. In the other words, 0 mmol L^−1^ of H^+^ was chosen as the proper concentration of hydrogen ions in the donor phase. This graph provides information about the impact of ion balance on the achieved peak areas. It is illustrated that the extraction recovery dramatically rises with ion balance decline. The lowest extraction output was acquired when the hydrogen ion concentration was equal in the donor and acceptor solutions. The current findings highlight the importance of the ion balance factor in EME performance.

#### 3.1.3. Time of Applied Potential

According to the theoretical viewpoint, EME is taken as an equilibrium method. This means that the extraction yield augments quickly against time within the initial extraction, and afterwards, after a duration, the system achieves equilibrium. The equilibrium period for EME is short due to applying potential [29]. A range of 10–20 min was tested as the BA extraction time (Figure 1). The concentration of extracted BAs in the acceptor solution dramatically increased from 10 to 20 min, whereas the GC-MS signals gradually diminished from 20 to 50 min, owing to pH variations in the donor and acceptor phases [26]. It was found that 20 min is the best time for the electro-migration of BAs across the SLM into HCl solution.

#### 3.1.4. Applied Potential

One of the most important factors involved in electro-membrane extraction efficiency is the applied potential magnitude. It is obvious that the mass transfer of ionized species via the SLM into the acceptor phase relies on the level of electric force [30]. To determine the effect of the voltage value on the extraction output, this factor was optimized and the peak area of GC-MS was considered as a function of the voltage level. In the current investigation, the extraction voltage varied between 20–100 V. The consequences demonstrated that raising the voltage up to 60 V gave rise to an enhancement in extraction recovery (Figure 1). However, the electric force above 60 V brought about bubble formation due to electrolysis and instability in EME system. Therefore, 60 V was selected as the optimal value.

### 3.2. Optimization of DLLME Process

#### 3.2.1. pH Level

For the optimization of the pH level in the sample solution after the EME process, the range of 2–11 was investigated. As shown in Figure 2, the analyte extraction rose slightly with pH augmentation. Based on the physicochemical attributes of BAs and the high level of pKa, the highest signal from the GC/MS was observed at pH 11. Thus, the optimum pH value, as one of the major variables in the DLLME method, was 11.

#### 3.2.2. Dispersive Solvent Volume

Due to the selection of dispersive solvent, methanol, acetonitrile and ethanol were assessed. The observations proved that acetonitrile could excellently distribute extraction solvent droplets throughout the aqueous solution and formed a cloudy state. As a matter of fact, acetonitrile acted as a mediator between the extraction phase and sample solution. In an attempt to optimize the amount of disperser solvent, 300, 475, 625 and 800 μL of acetonitrile were examined. Figure 2 indicates that the highest extraction output was attained at 625 μL of acetonitrile. The lower amount was not able to form a cloudy solution, and accordingly the extraction of BAs was not entirely accomplished. Contrary to expectations, the higher concentration of acetonitrile had a negative effect on extraction recovery due to a dilution effect.

#### 3.2.3. NaCl Percent (%)

To determine the impact of salt amount on extraction efficiency, 0–20% of NaCl was studied (Figure 2), and we compared the achieved responses. The ionic form of NaCl accumulated around water molecules in the aqueous solution. Thus, BAs could easily migrate to the extraction solvent. This phenomenon, called salting-out, occurred at 10% of NaCl. The salt molecules could not entirely be solved in the sample solution, owing to the saturated state. Therefore, it was concluded that a higher amount of salt reduces the response, and 10% was considered as the optimal value.

#### 3.2.4. Extraction Solvent Volume

According to former investigations, 1-octanol was selected as the suitable extraction solvent. This organic solvent has a low density and lower toxicity compared with other chlorinated organic solvents. This solvent is also compatible with GS/MS [23,31]. As is indicated in Figure 2, four levels of 1-octanol volume (40, 60, 80, 100) were tested. The bar chart represents that the highest peak area is related to 60 μL of 1-octanol. The lower volume of 1-octanol was not sufficient for the complete extraction of BAs. A gradual decline in extraction recovery was also observed at higher volumes Also, the dilution effect and reducing the enrichment factor were happened.

### 3.3. Method Validation

After the optimization step, method validation was undertaken for the verification of method reliability. Figures of merit are summarized in Table 1 for this purpose. The special concentrations of BAs were applied for the designation of calibration curve linearity. Good linearity, R^2^ > 0.97, was gained for the calibration curve. The relative standard deviation (RSD) was calculated for repeatability assessment and the confirmation of method precision. This value was attained between 4.9 and 7.0%. The recoveries were also between 94 and 98, which showed method accuracy. The ratio of BA concentration in the extraction solvent and sample solution was considered as the enrichment factor. This factor was between 36 and 41. The limit of detection (LOD) and limit of quantification (LOQ) were considered as three- and ten-times higher than background noise, respectively. The LODs were achieved in the range of 0.92 and 0.98, and LOQs were also obtained between 3.03 and 3.23. Table 2 represents a comparison between the current method and previous methods for analyzing BAs in beer samples. Different analytical techniques, such as GC/MS, were established for the detection and quantification of extracted BAs. Classical clean-up and extraction methods and a microextraction procedure were also used for the extraction of BAs. The recovery of the developed method in the present survey is more than 94%, while the other methods showed lower recovery. The enrichment factors were not reported for former methods, whereas the enrichment factor of HF-EME-DLLME was 36–41. Although electro-membrane extraction has gained remarkable attention due to its advantages, this procedure has major drawbacks. This analytical method has rather poor repeatability. Leakage might occur due to the porosity of the hollow fiber, especially when HCl solution is used as an acceptor phase. Additionally, obtaining the same volume of the acceptor phase for each injection is difficult, and that DLLME rectifies this problem. The combination of DLLME with electro-membrane extraction could facilitate the transfer of extracted analytes from the aqueous phase (HCl solution) to the organic phase (1-octanol) for GC/MC analysis and also increase the enrichment factor.

### 3.4. Real Sample Analysis

The applicability of the suggested method was validated by the analysis of ten non-alcoholic beer samples under optimum circumstances. Table 3 indicates the concentration of BAs detected in beer samples. Putrescine and tyramine, at concentrations of 3.87 and 2.33 µg g^−1^, were the BAs found in the highest amounts in non-alcoholic beers, respectively. A high intake of putrescine may increases the risk of colorectal adenocarcinoma [34]. Tyramine is formed in food products by the decarboxylation of tyrosine and triggers severe migraines in individuals [35]. The concentration ranges announced for tyramine, histamine, cadaverine and putrescine in this experiment were 0.86–2.33 µg g^−1^, 0.39–0.76 µg g^−1^, 0.52–0.93 µg g^−1^ and 0.89–3.87 µg g^−1^, respectively. The results manifest that this method is capable of the successful determination of four serious BAs in real beer samples. Figure 3 depicts the chromatograms of spiked and non-spiked beer samples. Separated peaks of target analytes without matrix interferences and with good baselines were observed.

## 4. Conclusions

There is a large volume of published studies describing the detriments of the long-term consumption of BA-containing food and beverages. Therefore, many researchers have focused on the determination of BAs in foodstuffs with several analytical methods. The current findings add to a growing body of literature on HF-EME and DLLME adequacy for the successful extraction of BAs and measuring BAs using GC/MS as a robust chromatographic instrument. The HF-EME process is fundamentally based on the electromigration of target analytes. Coupling this technique with DLLME could noticeably reduce the application of toxic organic solvents and boost extraction performance. The influential factors, which control the extraction mechanism, were optimized, and a verified method was employed for the measurement of putrescine, histamine, cadaverine and tyramine in non-alcoholic beers. The popularity of non-alcoholic beers has increased due to the lack of alcohol and their side effects on human well-being. The global non-alcoholic beer market has experienced an upward trend in recent years. Therefore, the analysis of BAs in this kind of beverage is obligatory. It has been established that the developed analytical procedure could successfully determine hazardous BAs in real non-alcoholic beer samples. Nevertheless, to develop a full picture of the suggested analytical method, additional surveys will be needed in the future.

## Figures and Tables

**Figure 1 foods-12-01141-f001:**
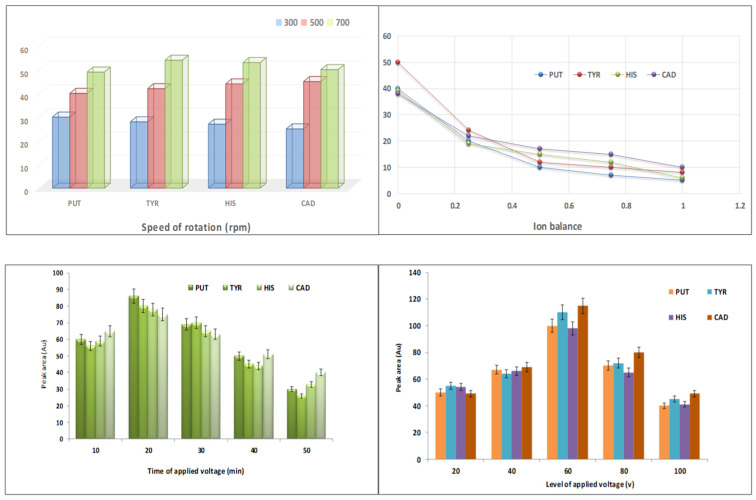
Optimization of speed of rotation, ion balance, time of applied voltage and level of applied voltage (HF-EME stage).

**Figure 2 foods-12-01141-f002:**
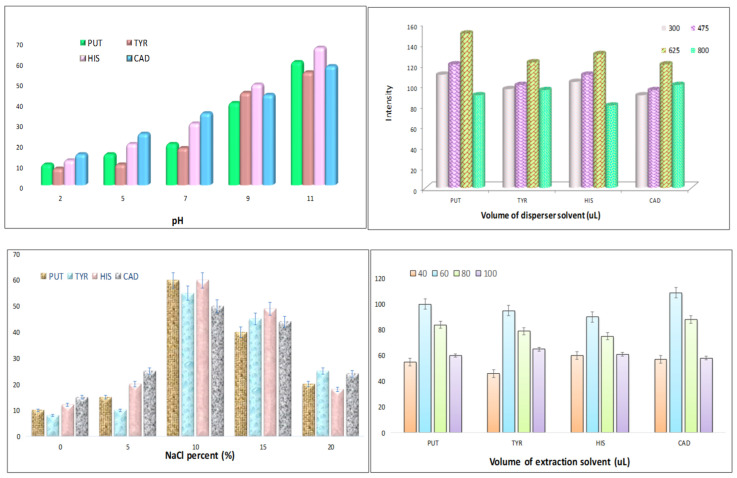
Optimum pH, volume of dispersive solvent, NaCl percentage and volume of extraction solvent (micro-DLLME stage).

**Figure 3 foods-12-01141-f003:**
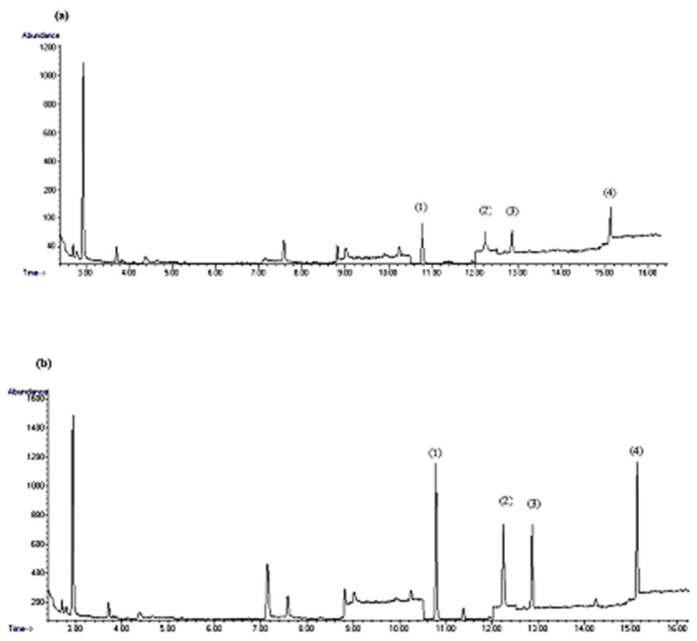
The chromatogram of (**a**) non-spiked and (**b**) spiked sample at a concentration 1000 ng g^−1^ of BAs. Putrescine (1), cadaverine (2), histamine (3) and tyramine (4) by the proposed method under optimum conditions.

**Table 1 foods-12-01141-t001:** Figures of merit of the proposed technique to measure BAs in non-alcoholic beer samples.

BAs	DLR (ng g^−1^)	R^2^	LOD (ng g^−1^)	LOQ (ng g^−1^)	RSD %	Recovery (%)	EF
HYS	1–5000	0.9906	0.92	3.03	7.0	96	41
CAD	1–5000	0.9760	0.98	3.23	5.5	97	37
TYR	1–5000	0.9882	0.93	3.06	4.9	94	36
PUT	1–5000	0.9958	0.96	3.16	6.2	98	39

**Table 2 foods-12-01141-t002:** Comparison of suggested method and previous procedures for BA extraction and quantification in beer samples.

Instrumentation	Extraction Method	Food Products	Linear Range (ng g^−1^)	Recovery (%)	LOD	LOQ	RSD (%)	R^2^	EF	Literature
GC/MS ^a^	DLLME ^b^	Beer	-	73–95	0.3–2.9 μg L^−1^	1–9.5 μg L^−1^	0.3–2.9	1–6	-	[32]
Capillary electrophoresis/UV	LLE ^c^	Beer	-	70.4–119.6	0.36–3.67 μmol L^−1^	1.2–12.2 μmol L^−1^	4.7<	-	-	[16]
LC/HRMS ^d^	MSPE ^e^	Beer	-	32.9–83.5	0.02–0.05 μg L^−1^	0.05–0.1 μg L^−1^	2.2–5.0	-	-	[33]
GC/MS	HF-EME-DLLME ^f^	Non-alcoholic beer	1–5000	94–98	0.92–0.98	3.03–3.23	4.9–7.0	0.9760–0.9958	36–41	Current study

^a^: Gas chromatography/mass spectrometry, ^b^: Dispersive liquid liquid microextraction, ^c^: Liquid liquid extraction, ^d^: Liquid chromatography/high resolution mass spectrometry, ^e^: Magnetic solid phase extraction, ^f^: Proposed method.

**Table 3 foods-12-01141-t003:** Analytical results of BAs in non-alcoholic beer samples using the offered method.

Samples	PUT			CAD			HYS			TYR		
	Found Concentration (µg g^−1^)	Added Amount (µg g^−1^)	Analyzed Amount (µg g^−1^)	Found Concentration (µg g^−1^)	Added Amount (µg g^−1^)	Analyzed Amount (µg g^−1^)	Found Concentration (µg g^−1^)	Added Amount (µg g^−1^)	Analyzed Amount (µg g^−1^)	Found Concentration (µg g^−1^)	Added Amount (µg g^−1^)	Analyzed Amount (µg g^−1^)
1	2.10 ± 0.13 ^a^	2	4.01 ± 0.24	0.80 ± 0.04	2	2.71 ± 0.14	0.76 ± 0.05	2	2.64 ± 0.18	1.07 ± 0.05	2	2.88 ± 0.14
2	1.84 ± 0.11	2	3.76 ± 11.65	0.65 ± 0.03	2	2.57 ± 0.14	0.54 ± 0.03	2	2.43 ± 0.17	0.86 ± 0.04	2	2.68 ± 0.13
3	0.99 ± 0.06	2	2.93 ± 0.18	0.93 ± 0.05	2	2.84 ± 0.15	0.49 ± 0.03	2	2.39 ± 0.16	1.43 ± 0.07	2	3.22 ± 0.15
4	2.31 ± 0.14	2	4.22 ± 0.26	0.52 ± 0.02	2	2.44 ± 0.13	0.41 ± 0.02	2	2.31 ± 0.16	1.15 ± 0.05	2	2.96 ± 0.14
5	1.70 ± 0.10	2	3.62 ± 0.22	0.76 ± 0.04	2	2.67 ± 0.14	0.50 ± 0.03	2	2.40 ± 0.16	0.87 ± 0.04	2	2.69 ± 0.13
6	0.89 ± 0.05	2	2.83 ± 0.17	0.61 ± 0.03	2	2.53 ± 0.13	0.48 ± 0.03	2	2.38 ± 0.16	2.33 ± 0.11	2	4.07 ± 0.19
7	3.0 ± 0.18	2	4.90 ± 0.30	0.57 ± 0.03	2	2.49 ± 0.13	0.39 ± 0.02	2	2.29 ± 0.16	0.90 ± 0.04	2	2.72 ± 0.13
8	2.44 ± 0.15	2	4.35 ± 0.26	0.68 ± 0.03	2	2.59 ± 0.14	0.60 ± 0.04	2	2.49 ± 0.17	1.19 ± 0.05	2	2.99 ± 0.14
9	0.92 ± 0.05	2	2.86 ± 0.17	0.72 ± 0.03	2	2.63 ± 0.14	0.49 ± 0.03	2	2.39 ± 0.16	1.23 ± 0.06	2	3.03 ± 0.14
10	3.87 ± 0.23	2	5.75 ± 0.35	0.69 ± 0.03	2	2.60 ± 0.14	0.39 ± 0.02	2	2.29 ± 0.16	1.06 ± 0.05	2	4.03 ± 0.19

^a^: Mean value ± std.

## Data Availability

No new data were created or analyzed in this study. Data sharing is not applicable to this article.

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
