# Peer review of "The Measurement of Hazardous Biogenic Amines in Non-Alcoholic Beers: Efficient and Applicable Miniaturized Electro-Membrane Extraction Joined to Gas Chromatography-Mass Spectrometry"

_foods, 2023, doi:10.3390/foods12061141_

Round 1

Reviewer 1 Report

Biogenic amines have been preconcentrated  using electromembrane extraction (EME) in other foods but the combination of EME and dispersive liquid liquid extraction (DLLME) seems uncommon.  Although beer samples were not analyzed, the following references are relevant and should be cited and discussed. The reproducibility of EME was cited as an issue (line 319) but no details were given.  The RSD in Table 3 was a little worse but not unreasonable. Does adding DLLME minimize some to the RSD with respect to EME? What would the RSD be without DLLME?

Conductive Cloth-Assisted Electromediated Extraction for the Determination of Biogenic Amines from Beverages

M Amayreh, C Basheer, A Hassan - Arabian Journal for Science and …, 2022 - Springer … electromediated extraction to preconcentrate biogenic amines (… The extraction variables
influencing the recoveries of BAs from … The efficiency of 50 V in electromembrane extraction has …

TML] … -assisted three-phase liquid–liquid extraction coupled with liquid chromatography–mass spectrometry for the determination of eight biogenic amines in foods

T Zhou, A Yigaimu, T Muhammad, P Jian, L Sha… - Food Chemistry, 2022 - Elsevier … extraction and back extraction in one step and one pot with a regular sample volume. Another
objective of the work was to simultaneously extract … -mediated electromembrane extraction …

Author Response

Thank you so much for the precious time and generous efforts you make to improve the manuscripts. My coauthors and I are indebted to the anonymous referees of this journal for their valuable comments. We have tried to change the manuscript according to your valuable opinions and we hope we have fulfilled your requests.

Dear Reviewer 1:

Thank you for the useful comments, suggestions and the time spent for the manuscript review.

Biogenic amines have been preconcentrated using electromembrane extraction (EME) in other foods but the combination of EME and dispersive liquid liquid extraction (DLLME) seems uncommon.

  • As you correctly mentioned, biogenic amines have been preconcentrated using electromembrane extraction (EME) in other foods. In this research, for the first time, the combination of EME and micro-DLLME has been employed to analyze TYR, HYS, CAD and PUT in beer samples. As you know, direct injection of aqueous acceptor phase (EME) into GC-MS is not possible. The combination of EME and micro-DLLME is so easy and fast and cause to open hand to select the wind ranges of compounds with different physicochemical properties. Also, this combination cause to improve the sensitivity, selectivity, separation and efficiency. As be reported in the text, the results were confirmed by standard addition test and the merit figures were acceptable.

Although beer samples were not analyzed, the following references are relevant and should be cited and discussed. Conductive Cloth-Assisted Electromediated Extraction for the Determination of Biogenic Amines from Beverages

Ref: M Amayreh, C Basheer, A Hassan - Arabian Journal for Science and …, 2022 - Springer … electromediated extraction to preconcentrate biogenic amines (… The extraction variables influencing the recoveries of BAs from … The efficiency of 50 V in electromembrane extraction has …

Ref: TML] … -assisted three-phase liquid–liquid extraction coupled with liquid chromatography–mass spectrometry for the determination of eight biogenic amines in foods T Zhou, A Yigaimu, T Muhammad, P Jian, L Sha… - Food Chemistry, 2022 - Elsevier … extraction and back extraction in one step and one pot with a regular sample volume. Another objective of the work was to simultaneously extract … -mediated electromembrane extraction …

  • As you correctly mentioned, the references were added in the text.

The reproducibility of EME was cited as an issue (line 319) but no details were given.

  • As you correctly mentioned, were added

The RSD in Table 3 was a little worse but not unreasonable. Does adding DLLME minimize some to the RSD with respect to EME? What would the RSD be without DLLME?

  • Absolutely yes, joining DLLME and EME improves RSD; getting the same volume of acceptor phase for each injection in EME process is hard that DLLME covers this problem.

 Dear Reviewer 2:

Thank you for the useful comments, suggestions and the time spent for the manuscript review.

Dears, unfortunately, as follows, some aspects must be improved by means of modifications/explanations.

Line 43; change ‘is’ with ‘are’.

  • As you correctly mentioned, was corrected.

Line 43; change ‘such’ with ‘such as’.

  • As you correctly mentioned, was corrected.

Line 45; change ‘divide to’ with ‘divide into’.

  • As you correctly mentioned, was corrected.

Line 49; change ‘trigger allergic’ with ‘triggers allergic’.

  • As you correctly mentioned, was corrected.

Line 51; change ‘consider as ‘with ‘considered as’.

  • As you correctly mentioned, was corrected.

Line 52-53; ‘The exogenous BAs uptake through daily diet  including, fermented food, meat products, cheese, fishery products, fruits, vegetables and  beverages’.What does it mean?

  • Thanks for your attention. This sentence was edited in the manuscript.

Line 52-53; ‘beers develops’?

  • The sentence was edited in the manuscript.

Line 63-64; ‘While the presence of BAs in this kind of beer is the serious issue in the beverages industry’?

  • The sentence was edited in the manuscript.

 Line 65; change ‘It was pointed that’ with ‘It was pointed out that’.

  • As you correctly mentioned, was corrected.

Line 67; change  ‘by microorganisms exist’  with  ‘by microorganisms existing’.

  • As you correctly mentioned, was corrected.

Line 74; change ‘made to analysis of Bas’with ‘made to the analysis of Bas’.

  • As you correctly mentioned, was corrected.

Line 90; change ‘factor are potentiated’ with  ‘factor is potentiated’.

  • As you correctly mentioned, was corrected.

Line 91; change  ‘subsets of LPME could efficiently extract various analytes from food’ with.  ‘subsets of LPME that could efficiently extracts various analytes from food’.

  • As you correctly mentioned, was corrected.

Line 91-92; ‘In this method, the trivial microliters of extraction solvent and disperser solvent swiftly injected to aqueous sample solution.’?

  • As you correctly mentioned, the sentence was edited.

Line 93; ‘scatters’?  ‘enhance’?

  • As you correctly mentioned, the sentence was edited.

 Line 98; ‘is based on electromigration’?

  • As you correctly mentioned, the sentence was edited.

Line 99; ‘are transferred from’?

  • As you correctly mentioned, the sentence was edited.

Line 99; ‘was designed to’?

  • As you correctly mentioned, the sentence was edited.

Line 105; ‘ ‘which is considered’?

  • As you correctly mentioned, the sentence was edited.

 Line 108; ‘contributes to potentiation’? 

  • As you correctly mentioned, the sentence was edited.

 Line 109; ‘improves’?

  • As you correctly mentioned, the sentence was edited.

 Line 163; ‘were poured’?

  • As you correctly mentioned, the sentence was edited.

 Line 165; ‘were added’?

  • As you correctly mentioned, the sentence was edited.

Statistical analysis: It is not clear neither the way or the statistical tests utilized for analyzing the comparisons. Even, number of replicates and factors are not reported and described. Level of significance are absent. 

  • As you correctly mentioned, were added.

 Line 187; change ‘affected on both extraction methods’ with ‘affected both extraction methods’.

  • As you correctly mentioned, the sentence was edited.

 Line 188; ‘were optimized by one variable’?

  • As you correctly mentioned, the sentence was edited.

 Line 194; ‘the optimization response’?

  • As you correctly mentioned, was corrected.

Line 199; ‘Take the following variables into account could extensively improve expected results’ check the meaning of the sentence.

  • As you correctly mentioned, was corrected.

Line 211: ‘agitation decreases the’?

  • As you correctly mentioned, the sentence was edited.

Line 213; ‘, three levels of rotation’?

  • As you correctly mentioned, the sentence was edited.

Dear Reviewer 3:

Thank you for the useful comments, suggestions and the time spent for the manuscript review.

 The article deals with the investigation of the combination of hollow fiber-electromembrane extraction (HF-EME) and dispersive liquid-liquid microextraction (DLLME) followed by gas chromatography mass spectrometry (GC/MS) for the analysis of histamine, putrescine, tyramine, cadaverine in nonalcoholic beers. The topic is good. The manuscript has been generally well designed and written. However, it needs some revision. My comments and questions are below;

 -       Line 15: Iran?

  • Yes, this project was done as team working in different universities of Iran

-       Abstract: Some information about the method validation parameters should be added to the abstract!

  • The fundamental merit figures for method validation were added to the abstract.

-       Since this study is a method development study, the usage purposes of the chemicals used in the analysis should be given.

  • As you correctly mentioned, the usage purposes of the chemicals were added to the manuscript (Initial sample preparation section). The usage purposes of 2-nitrophenyl octyle ether (NPOE) as supported liquid membrane in the electromembrane process was completely explained in the introduction and result and discussion part. The usage reasons of acetonitrile, 1-octanol and NaCl in the dispersive liquid-liquid membrane were also explained in the result and discussion part.

-       Please give the statistical model design of your study!

  • As you correctly mentioned was added in "statistical analysis" section.

-       What about legal limits for these BAs? The researchers should have discussed this!

  • The total safe dose for BAs was pointed out in the first paragraph of the introduction and highlighted. Whereas the particular legal limit for the beers have not been reported by related organizations.

Thank you again for your precious comments. We tried to correct and/or explain completely the whole comments. It would be our pleasure if you reconsider the resubmit manuscript and inform us about any other corrections necessary for the improvement of the manuscript.

Best regards

Reviewer 2 Report

  Dears, unfortunately, as follows, some aspects must be improved by means of modifications/explanations.

 Line  43; change ‘is’ with ‘are’.

Line  43; change ‘such ’ with ‘such as’.

Line  45; change ‘divide  to’ with ‘divide into’.

Line  49; change ‘trigger allergic’ with ‘triggers allergic’.

Line  51; change ‘consider as ‘with ‘considered as’.

Line  52-53; ‘The exogenous BAs uptake through daily diet  including, fermented food, meat products, cheese, fishery products, fruits, vegetables and  beverages’.

What does it mean?

Line  52-53; ‘beers develops’ ?

Line  63-64; ‘While the presence of BAs in this kind of beer is the serious issue in the beverages industry’?

Line  65; change ‘It was pointed that’ with ‘It was pointed out that’.

Line  67; change  ‘by microorganisms exist’  with  ‘by microorganisms existing’.

Line  74; change  ‘made to analysis of Bas’with ‘made to the analysis of Bas’.

Line  90; change  ‘factor are potentiated’ with  ‘factor is potentiated’.

Line  91; change  ‘subsets of LPME could efficiently extract various analytes from food’ with

  ‘subsets of LPME that could efficiently extracts various analytes from food’.

Line  91-92; ‘In this method, the 91 trivial microliters of extraction solvent and disperser solvent swiftly injected to aqueous 92 sample solution.’?

Line  93; ‘scatters’ ?  ‘enhance’ ?

Line  98; ‘is based on electromigration’?

Line  99; ‘are transferred from’ ?

Line  99; ‘was designed to’ ?

Line  105; ‘ ‘which is considered’ ?

Line  108; ‘contributes to potentiation’ ?   

Line  109; ‘improves’ ?

Line  163; ‘were poured’ ?

Line  165; ‘were added’ ?

Statistical analysis:

It is not clear neither the way or the statistical tests utilized for analyzing the comparisons. Even, number of replicates and factors are  not reported and described. Level of significance are absent.     

Line  187; change ‘affected on both extraction methods’ with ‘affected both extraction  methods’ .

Line  188; ‘were optimized by one variable’?

Line  194; ‘the optimization response’ ?

Line  199; ‘Take the following variables into account could extensively improve expected 199 results’ check the meaning of the sentence.

Line  211: ‘agitation decreases the’ ?

Line  213; ‘, three levels of rotation’ ?

Author Response

(The authors gave the same response as above.)

Reviewer 3 Report

Foods

foods-2236136

The measurement of hazardous biogenic amines in non-alcoholic beers: Efficient and applicable miniaturized electro-membrane extraction joined to gas chromatography-mass spectrometry

Dear Editor,

The article deals with the investigation of the combination of hollow fiber-electromembrane extraction (HF-EME) and dispersive liquid-liquid microextraction (DLLME) followed by gas chromatography mass spectrometry (GC/MS) for the analysis of histamine, putrescine, tyramine, cadaverine in nonalcoholic beers. The topic is good. The manuscript has been generally well designed and written. However, it needs some revision. My comments and questions are below;

-       Line 15: Iran?

-       Abstract: Some information about the method validation parameters should be added to the abstract!

-       Since this study is a method development study, the usage purposes of the chemicals used in the analysis should be given.

-       Please give the statistical model design of your study!

-       What about legal limits for these BAs? The researchers should have discussed this!

Author Response

(The authors gave the same response as above.)

Round 2

Reviewer 2 Report

It should be better to specify if a one-way ANOVA has been utilized as a statistical system for testing differences.